# DATASET AUGMENTATION IN FEATURE SPACE

**Terrance DeVries and Graham W. Taylor**
School of Engineering
University of Guelph
Guelph, ON N1G 2W1, Canada
`{terrance,gwtaylor}@uoguelph.ca`

## ABSTRACT

Dataset augmentation, the practice of applying a wide array of domain-specific transformations to synthetically expand a training set, is a standard tool in supervised learning. While effective in tasks such as visual recognition, the set of transformations must be carefully designed, implemented, and tested for every new domain, limiting its re-use and generality. In this paper, we adopt a simpler, domain-agnostic approach to dataset augmentation. We start with existing data points and apply simple transformations such as adding noise, interpolating, or extrapolating between them. Our main insight is to perform the transformation not in input space, but in a learned feature space. A re-kindling of interest in unsupervised representation learning makes this technique timely and more effective. It is a simple proposal, but to-date one that has not been tested empirically. Working in the space of context vectors generated by sequence-to-sequence models, we demonstrate a technique that is effective for both static and sequential data.

## 1 INTRODUCTION

One of the major catalysts for the resurgence of neural networks as "deep learning" was the influx of the availability of data. Labeled data is crucial for any supervised machine learning algorithm to work, even moreso for deep architectures which are easily susceptible to overfitting. Deep learning has flourished in a few domains (e.g. images, speech, text) where labeled data has been relatively simple to acquire. Unfortunately most of the data that is readily available is unstructured and unlabeled and this has prevented recent successes from propagating to other domains. In order to leverage the power of supervised learning, data must be manually labeled, a process which requires investment of human effort. An alternative to labeling unlabeled data is to generate new data with known labels. One variant of this approach is to create synthetic data from a simulation such as a computer graphics engine (Shotton et al., 2013; Richter et al., 2016), however, this may not work if the simulation is not a good representation of the real world domain. Another option is dataset augmentation, wherein the existing data is transformed in some way to create new data that appears to come from the same (conditional) data generating distribution (Bengio et al., 2011). The main challenge with such an approach is that domain expertise is required to ensure that the newly generated data respects valid transformations (i.e. those that would occur naturally in that domain).

In this work, we consider augmentation not by a domain-specific transformation, but by perturbing, interpolating, or extrapolating between existing examples. However, we choose to operate not in input space, but in a learned feature space. Bengio et al. (2013) and Ozair & Bengio (2014) claimed that higher level representations expand the relative volume of plausible data points within the feature space, conversely shrinking the space allocated for unlikely data points. As such, when traversing along the manifold it is more likely to encounter realistic samples in feature space than compared to input space. Unsupervised representation learning models offer a convenient way of learning useful feature spaces for exploring such transformations. Recently, there has been a return to interest in such techniques, leading to, e.g., variational autoencoders (Kingma & Welling, 2014), generative adversarial networks (Goodfellow et al., 2014), and generative stochastic networks (Alain et al., 2016), each of which could be used to generate useful feature spaces for augmentation.

By manipulating the vector representation of data within a learned feature space a dataset can be augmented in a number of ways. One of the most basic transformations that can be applied to the

data is to simply add random noise to the context vector. In the context of class-imbalanced data, Chawla et al. (2002) proposed *interpolating* between samples in feature space. Similarly extrapolation between samples could also be applied. We investigate some of these methods to see which is most effective for improving the performance of supervised learning models when augmented data is added to the dataset.

In this work, we demonstrate that extrapolating between samples in feature space can be used to augment datasets and improve the performance of supervised learning algorithms. The main benefit of our approach is that it is domain-independent, requiring no specialized knowledge, and can therefore be applied to many different types of problems. We show that models trained on datasets that have been augmented using our technique outperform models trained only on data from the original dataset. Just as dataset augmentation in input space has become standard for visual recognition tasks, we recommend dataset augmentation in feature space as a domain-agnostic, general-purpose framework to improve generalization when limited labeled data is available.

## 2    RELATED WORK

For many years, dataset augmentation has been a standard regularization technique used to reduce overfitting while training supervised learning models. Data augmentation is particularly popular for visual recognition tasks as new data can be generated very easily by applying image manipulations such as shifting, scaling, rotation, and other affine transformations. When training LeNet5, one of the most early and well-known convolutional neural network architectures, LeCun et al. (1998) applied a series of transformations to the input images in order to improve the robustness of the model. Krizhevsky et al. (2012) also used image transformations to generate new data when training the renowned AlexNet model for the 2012 Large Scale Visual Recognition Challenge (ILSVRC). They claimed that dataset augmentation reduced the error rate of the model by over 1%. Creating new data has since been a crucial component of all recent large-scale image recognition models.

Unfortunately, dataset augmentation is not as straightforward to apply in all domains as it is for images. For example, Schlüter & Grill (2015) investigated a variety of data augmentation techniques for application to singing voice detection. These include adding Gaussian noise to the input, shifting the pitch of the audio signal, time stretching, varying the loudness of the audio signal, applying random frequency filters, and interpolating between samples in input space. They found that only pitch shifting and random frequency filtering appeared to improve model performance. While performing well on audio data, these augmentation techniques cannot be applied to other domains. As such, the process of designing, implementing, and evaluating new data augmentation techniques would need to be repeated for each new problem.

Important to our work are sequence-to-sequence learning (seq2seq) models which were first developed independently by Cho et al. (2014) and Sutskever et al. (2014). Generally these models convert a sequence of inputs from one domain into a fixed-length context vector which is then used to generate an output sequence, usually from a different domain. For example, the first application of seq2seq learning by Cho and Sutskever was to translate between English and French. Sequence-to-sequence learning has recently been used to achieve state-of-the-art results on a large variety of sequence learning tasks including image captioning (Vinyals et al., 2015b), video captioning (Venugopalan et al., 2015), speech recognition ((Chan et al., 2016), (Bahdanau et al., 2016)), machine translation ((Jean et al., 2015), (Luong et al., 2015)), text parsing (Vinyals et al., 2015a), and conversational modeling (Vinyals & Le, 2015). The seq2seq architecture can also be used to create sequence autoencoders (SA) by creating a model that learns to reconstruct input sequences in its output (Srivastava et al., 2015; Dai & Le, 2015). We use a variant of sequence autoencoders in our work to create a feature space within which we can manipulate data to augment a training set.

## 3    MODEL

Our dataset augmentation technique works by first learning a data representation and then applying transformations to samples mapped to that representation. Our hypothesis is that, due to manifold unfolding in feature space, simple transformations applied to encoded rather than raw inputs will result in more plausible synthetic data. While any number of representation learning models could

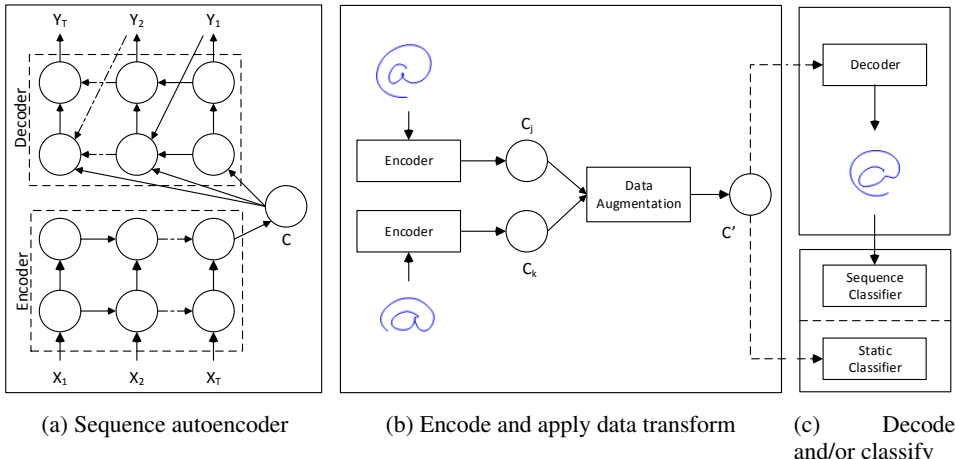

(a) Sequence autoencoder  (b) Encode and apply data transform  (c)  Decode and/or classify

Figure 1: System architecture composed of three steps. (a) A sequence autoencoder learns a feature space from unlabeled data, representing each sequence by a context vector ($C$). (b) Data is encoded to context vectors and augmented by adding noise, interpolating, or extrapolating (here we depict interpolation). (c) The resulting context vectors can either be used directly as features for supervised learning with a static classifier, or they can be decoded to reconstruct full sequences for training a sequence classifier.

be explored, we use a sequence autoencoder to construct a feature space. The main reason we adopt SA is that we favour a generic method that can be used for either time series or static data.

### 3.1 SEQUENCE AUTOENCODER

An autoencoder consists of two parts: an encoder and a decoder. The encoder receives data as input and, by applying one or more parametrized nonlinear transformations, converts it into a new representation, classically lower-dimensional than the original input. The decoder takes this representation and tries to reconstruct the original input, also by applying one or more nonlinear transformations. Various regularized forms of autoencoders have been proposed to learn *overcomplete* representations.

A sequence autoencoder works in a similar fashion as the standard autoencoder except that the encoder and decoder use one or more recurrent layers so that they can encode and decode variable-length sequences. In all of our experiments, we use a stacked LSTM (Li & Wu, 2015) with two layers for both the encoder and decoder (Figure 1a). During the forward pass, the hidden states of the recurrent layers are propagated through the layer stack. The encoder's hidden state at the final time step, called the *context vector*, is used to seed the hidden state of the decoder at its first time step.

The main difference between our implementation of the SA and that of Dai & Le (2015) is how the context vector is used in the decoder. Dai and Le follow the original seq2seq approach of Sutskever et al. (2014) and use the context vector as input to the decoder only on the first time step, then use the output of the previous times step as inputs for all subsequent time steps as follows:

$$\mathbf{y}_0 = f(\mathbf{s}_0, \mathbf{c})$$
$$\mathbf{y}_t = f(\mathbf{s}_{t-1}, \mathbf{y}_{t-1})$$

where $f$ is the LSTM function, $\mathbf{s}$ is the state of the LSTM (both hidden and cell state), $\mathbf{c}$ is the context vector, and $\mathbf{y}$ is the output of the decoder. We instead modify the above equation so that the decoder is conditioned on the context vector at each time step as was done in (Cho et al., 2014):

$$\mathbf{y}_0 = f(\mathbf{s}_0, \mathbf{c})$$
$$\mathbf{y}_t = f(\mathbf{s}_{t-1}, \mathbf{y}_{t-1}, \mathbf{c}).$$

We found that conditioning the decoder on the context vector each time step resulted in improved reconstructions, which we found to be critical to the success of the data augmentation process.

### 3.2 Augmentation in Feature Space

In order to augment a dataset, each example is projected into feature space by feeding it through the sequence encoder, extracting the resulting context vector, and then applying a transformation in feature space (Figure 1b). The simplest transform is to simply add noise to the context vectors, however, there is a possibility with this method that the resulting vector may not resemble the same class as the original, or even any of the known classes. In our experiments, we generate noise by drawing from a Gaussian distribution with zero mean and per-element standard deviation calculated across all context vectors in the dataset. We include a $\gamma$ parameter to globally scale the noise:

$$c_i' = c_i + \gamma X, \ X \sim \mathcal{N}\{0, \sigma_i^2\} \tag{1}$$

where $i$ indexes the elements of a context vector which corresponds to data points from the training set. A more directed approach for data augmentation follows the techniques introduced by Chawla et al. (2002). For each sample in the dataset, we find its $K$ nearest neighbours in feature space which share its class label. For each pair of neighbouring context vectors, a new context vector can then be generated using interpolation:

$$\mathbf{c}' = (\mathbf{c}_k - \mathbf{c}_j)\lambda + \mathbf{c}_j \tag{2}$$

where $\mathbf{c}'$ is the synthetic context vector, $\mathbf{c}_i$ and $\mathbf{c}_j$ are neighbouring context vectors, and $\lambda$ is a variable in the range $\{0, 1\}$ that controls the degree of interpolation. In our experiments, we use $\lambda = 0.5$ so that the new sample balances properties of both original samples. In a similar fashion, extrapolation can also be applied to the context vectors:

$$\mathbf{c}_j' = (\mathbf{c}_j - \mathbf{c}_k)\lambda + \mathbf{c}_j. \tag{3}$$

In the case of extrapolation, $\lambda$ is a value in the range $\{0, \infty\}$ which controls the degree of extrapolation. While $\lambda$ could be drawn from a random distribution for each new sample we found that setting $\lambda = 0.5$ worked well as a default value in most cases, so we use this setting in all of our tests.

Once new context vectors have been created, they can either be used directly as input for a learning task, or they can be decoded to generate new sequences (Figure 1c). When interpolating between two samples, the resulting decoded sequence is set to be the average length of the two inputs. When extrapolating between two samples the length of the new sequence is set to be the same as that of $c_j$.

## 4 Experiments

In all experiments, we trained a LSTM-based sequence autoencoder in order to learn a feature space from the available training examples. Each hidden layer, including the context vector, had the same number of hidden units and a dropout probability of $p = 0.2$. The autoencoders were trained using Adam (Kingma & Ba, 2015) with an initial learning rate of 0.001, which was reduced by half whenever no improvement was observed in the validation set for 10 epochs. Finally, we reversed the order of the input sequences as suggested by Sutskever et al. (2014). We found that reversing the order of input sequences caused the model to train faster and achieve better final solutions.

For all classification experiments where interpolation or extrapolation was applied to generate new samples, we applied the following procedure unless otherwise stated. For each sample in the dataset we found the 10 nearest in-class neighbours by searching in feature space. We then interpolated or extrapolated between each neighbour and the original sample to produce a synthetic example which was added to the augmented dataset. For all tests, the baseline model and the augmented dataset model(s) were trained for the same number of weight updates regardless of dataset size.

### 4.1 Visualization - Sinusoids

To gain an intuition of the method we start by working with a synthetic dataset of sinusoids. Sinusoids work well as a test case for this technique as they have a known behaviour and only two dimensions (amplitude and time), so we can easily observe the effects of the dataset augmentation process. To create a training set, sinusoids were generated with amplitude, frequency, and phase drawn from a uniform distribution.

For this toy problem, we trained a sequence autoencoder with 32 hidden units in each layer. We then applied different data augmentation strategies to observe the effects on the "synthetic" sinusoids.

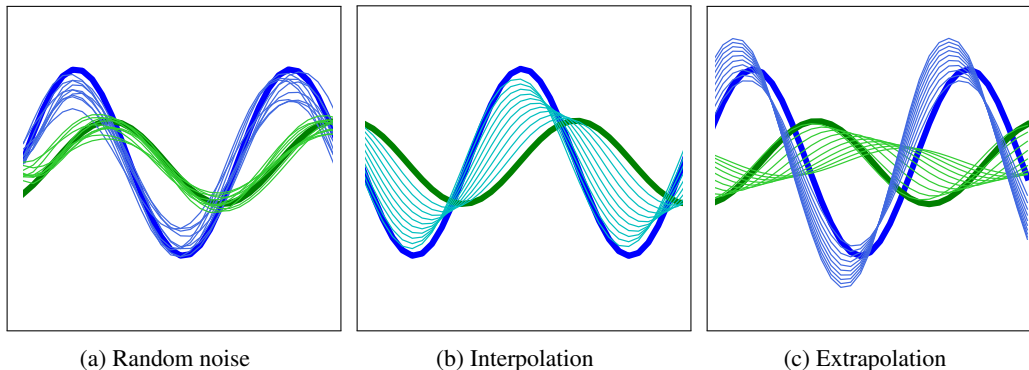

| (a) Random noise | (b) Interpolation | (c) Extrapolation |

Figure 2: Sinusoids with various transformations applied in feature space. (a) Random noise added with $\gamma = 0.5$. (b) Interpolation between two sinusoids for values of $\lambda$ between 0 and 1. (c) Extrapolation between two sinusoids for values of $\lambda$ between 0 and 1. Best viewed in colour.

For each test we extracted the context vectors of two input sinusoids, performed an operation, then decoded the resulting context vectors to generate new sequences.

We first augmented data by adding random noise to the context vectors before decoding. The noise magnitude parameter $\gamma$ from Equation 1 was set to 0.5. In Figure 2a the blue and green "parent" samples are shown in bold while the augmented "child" samples are thinner, lighter lines. Importantly, we observe that all new samples are valid sinusoids with stable, regular repeating patterns. Although mimicking the major properties of their parents the generated samples have small changes in amplitude, frequency, and phase, as would be the expected effect for the addition of random noise.

For a more directed form of data augmentation we experimented with interpolating between sinusoids within the space of the context vectors. Figure 2b demonstrates interpolation between two sinusoids using Equation 2 while varying the $\lambda$ parameter from 0 to 1. Unlike the results obtained by Bengio et al. (2013) where the transition between classes occurs very suddenly we find that the samples generated by our model smoothly transition between the two parent sinusoids. This is an exciting observation as it suggests that we can control characteristics of the generated samples by combining two samples which contain the desired properties.

In a similar fashion to interpolation we can also extrapolate between two samples using Equation 3. For this experiment we again vary the $\lambda$ parameter from 0 to 1 to generate a range of samples. As seen in Figure 2c, this appears to have the effect of exaggerating properties of each sinusoid with respect to the properties of the other sinusoid. For example, we see that new samples generated from the blue parent sinusoid increase in amplitude and decrease in phase shift. Conversely, samples generated from the green parent sinusoid decrease in amplitude and increase in phase shift. The behaviour of the extrapolation operation could prove very beneficial for data augmentation as it could be used to generate extra samples of rare or underrepresented cases within the dataset, which is a common failure case.

## 4.2 Visualization - UJI Pen Characters

The UJI Pen Characters dataset (v2) contains 11,640 instances of 97 different characters handwritten by 60 participants (Llorens et al., 2008). All samples were collected using a tablet PC and a stylus. Characters are defined by a sequence of X and Y coordinates, and include upper and lower case ASCII letters, Spanish non-ASCII letters, the 10 digits, and other common punctuation and symbols. As with the sinusoids in Section 4.1, handwritten characters are suitable for evaluating dataset augmentation methods as they have an expected shape and can be easily visualized.

As a preprocessing step for this dataset we first applied local normalization to each sample to get a fixed size, followed by a global normalization across the dataset as a whole. A sequence autoencoder with 128 hidden units per layer was trained to construct the feature space within which data augmentation could take place. Figure 3a demonstrates the effects of interpolating between characters in feature space. In this example we use the "@" symbol. We see that the resulting characters share

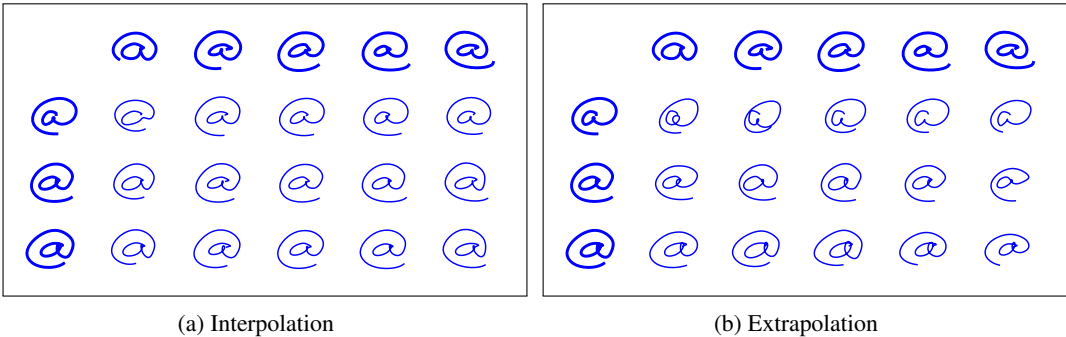

(a) Interpolation (b) Extrapolation

Figure 3: Interpolation (a) and extrapolation (b) between handwritten characters. Character (0,i) is interpolated/extrapolated with character (j,0) to form character (i,j), where i is the row number and j is the column number. Original characters are shown in bold.

characteristics of the two parent inputs, such as the length of the symbol's tail or the shape of the central "a". Visually the majority of generated samples appear very similar to their parents, which is expected from interpolation, but is not necessarily useful from the perspective of data augmentation.

When augmenting data for the purpose of improving performance of machine learning algorithms it is desirable to create samples that are different from the data that is already common in the dataset. To this end, extrapolating between samples is preferable, as shown in Figure 3b. Extrapolated data displays a wider variety compared to samples created by interpolation. We hypothesize that it is this added variability that is necessary in order for data augmentation to be useful.

### 4.3 SPOKEN ARABIC DIGITS

For our first quantitative test we use the Arabic Digits dataset (Lichman, 2013) which contains 8,800 samples of time series mel-frequency cepstrum coefficients (MFCCs) extracted from audio clips of spoken Arabic digits. Thirteen MFCCs are available for each time step in this dataset. To preprocess the data we apply global normalization. To evaluate our data augmentation techniques we used the official train/test split and trained ten models with different random weight initializations.

As a baseline model we trained a simple two layer MLP on the context vectors produced by a SA. Both models used 256 hidden units in each hidden layer. The MLP applied dropout with $p = 0.5$ after each dense layer. To evaluate the usefulness of different data augmentation techniques we trained a new baseline model on datasets that had been augmented with newly created samples. The techniques we evaluated were: adding random noise to context vectors, interpolating between two random context vectors from the same class, interpolating between context vectors and their nearest neighbours from the same class, and extrapolating between context vectors and their nearest neighbours from the same class. The results of our tests are summarized in Table 1.

Table 1: Test set error on Arabic Digits dataset averaged over 10 runs

| Model | Test Error (%) |
| --- | --- |
| Baseline (ours) | $1.36 \pm 0.15$ |
| Baseline + random noise (ours) | $1.10 \pm 0.15$ |
| Baseline + random interpolation (ours) | $1.82 \pm 0.21$ |
| Baseline + nearest neighbour interpolation (ours) | $1.57 \pm 0.19$ |
| Baseline + nearest neighbour extrapolation (ours) | $\mathbf{0.74 \pm 0.11}$ |
| (Hammami et al., 2012) | $\mathbf{0.69}$ |

We find that our simple baseline model achieves competitive performance after training on the extracted context vectors, demonstrating the feature extracting capability of the sequence autoencoder. The naïve data augmentation approach of adding random noise to the context vectors further improves performance. Of interest, we find that adding new samples generated using interpolation techniques diminishes the performance of the model, which confirms our hypothesis that good data augmentation techniques should add variability to the dataset. Of the two interpolation techniques,

we see that interpolating between neighbouring samples performs better than simply interpolating with randomly chosen samples of the same class. Finally we observe that extrapolating between samples improves model performance significantly, reducing the baseline error rate by almost half. Our results rival those of Hammami et al. (2012), which to our knowledge are state-of-the-art on this dataset.

## 4.4 AUSTRALIAN SIGN LANGUAGE SIGNS (AUSLAN)

Our second quantitative test was conducted on the Australian Sign Language Signs dataset (AUS-LAN). AUSLAN was produced by Kadous (2002) and contains 2,565 samples of a native signer signing 95 different words or phrases while wearing high quality position tracking gloves. Each time series sample is, on average, 57 frames in length and includes 22 features: roll, pitch, yaw, finger bend, and the 3D coordinates of each hand. To preprocess the raw data we first locally centre each sample and then apply global normalization. For evaluation, we perform cross validation with 5 folds, as is common practice for the AUSLAN dataset.

The baseline model for these tests was a two layer MLP with 512 hidden units in each layer, with dropout ($p = 0.5$) applied on each. Similar to Arabic Digits, dataset we find that the simple MLP can achieve competitive results when trained on the context vectors extracted from the sequence autoencoder (see Table 2). In this case, however, we observe that adding random noise to the context vectors did not improve performance. One possible explanation for this outcome is that the AUS-LAN dataset has much more classes than the Arabic Digits dataset (95 versus 10) so there is higher probability of a randomly augmented context vector jumping from one class manifold to another. Traversing instead along the representational manifold in a directed manner by extrapolating between neighbouring samples results in improved performance over that of the baseline model. Our results also match the performance of Rodríguez et al. (2005), which to our knowledge is the best 5-fold cross validation result for the AUSLAN dataset.

Table 2: CV error on AUSLAN dataset averaged over 5 folds

| Model | Test Error (%) |
|---|---|
| Baseline (ours) | $1.53 \pm 0.26$ |
| Baseline + random noise (ours) | $1.67 \pm 0.12$ |
| Baseline + interpolation (ours) | $1.87 \pm 0.44$ |
| Baseline + extrapolation (ours) | $\mathbf{1.21 \pm 0.26}$ |
| (Rodríguez et al., 2005) | $\mathbf{1.28}$ |

## 4.5 UCFKINECT

The final time series dataset we considered was the UCF Kinect action recognition dataset (Ellis et al., 2013). It contains motion capture data of participants performing 16 different actions such as *run, kick, punch,* and *hop*. The motion capture data consists of 3-dimensional coordinates for 15 skeleton joints for a total of 45 attributes per frame. In total there are 1,280 samples within the dataset. To preprocess the dataset we first shift the coordinates of each sample so that the central shoulder joint of the first frame is located at the origin. Global normalization is also applied.

With the UCFKinect dataset our main goal was to determine the effectiveness of interpolation in feature space for generating new sequences that combine the characteristics and actions of the two "seed" examples. We found that in order to produce natural looking results, the two actions to be combined must already share some properties. For example, Figure 4a and 4b show motion capture sequences of a person stepping forward and a person stepping to the left, respectively. Both of these actions take approximately the same amount of time to perform, and each skeleton moves their left leg first, then their right leg. Due to these preexisting similarities the action sequences can be interpolated in feature space to produce a natural looking sequence of a skeleton stepping diagonally forward and to the left (Figure 4c). These results emulate what was previously observed in Section 4.3, which indicated that similar properties are necessary for successful blending of examples.

Our secondary goal with the UCFKinect dataset was to quantitatively evaluate the performance of extrapolation-based data augmentation. To compare to previous results, we used 4-fold cross validation (see Table 3 for a summary of results). We found that extrapolating between samples in

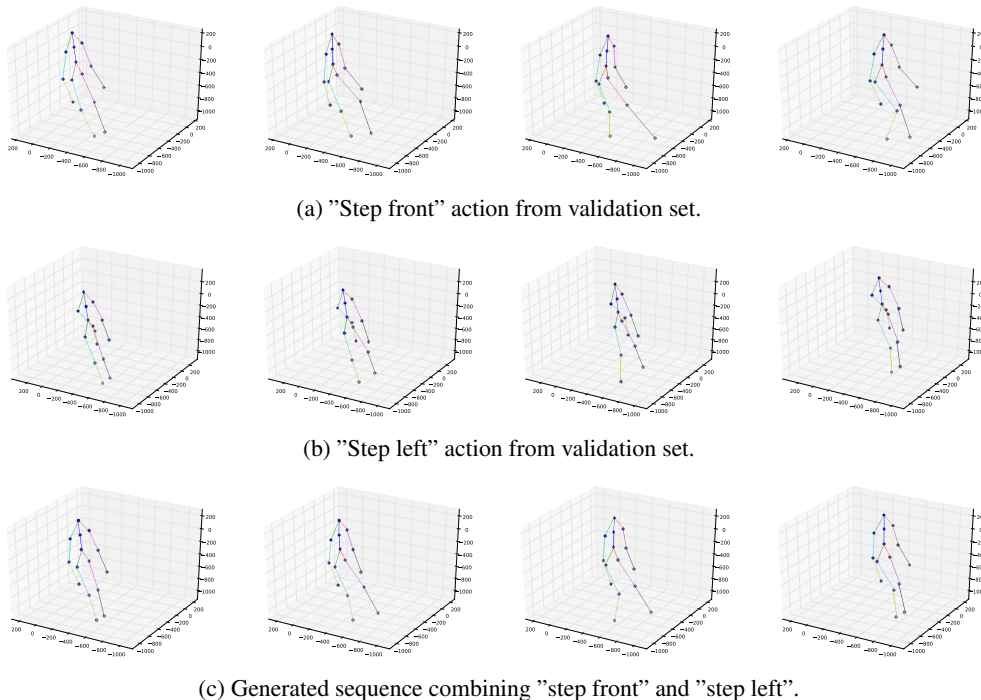

(a) "Step front" action from validation set.

(b) "Step left" action from validation set.

(c) Generated sequence combining "step front" and "step left".

Figure 4: A new motion capture sequence can be generated by interpolating between samples. By combining the "step front" action (a) with the "step left" action (b) we can generate a new sequence of a character stepping diagonally forward and the to left (c).

representational space improved the performance of our untuned model by more than 1%, which is quite significant. Our results are 2.5 percentage points below the current state-of-the-art result produced by Beh et al. (2014), but further tuning of the model could improve results.

Table 3: CV error on UCFKinect dataset averaged over 4 folds

| Model | Test Error (%) |
|---|---|
| Baseline (ours) | $4.92 \pm 2.09$ |
| Baseline + extrapolation (ours) | $3.59 \pm 1.61$ |
| (Beh et al., 2014) | **1.10** |

## 4.6 IMAGE CLASSIFICATION: MNIST AND CIFAR-10

Having successfully applied dataset augmentation in feature space to improve the accuracy of sequence classification tasks, we now experiment with applying our technique to static data. For these experiments we concentrate on the image domain where manual data augmentation is already prevalent. We find that augmenting datasets by extrapolating within a learned feature space improves classification accuracy compared to no data augmentation, and in some cases surpasses traditional (manual) augmentation in input space.

In our experiments we consider two commonly used small-scale image datasets: MNIST and CIFAR-10. MNIST consists of 28×28 greyscale images containing handwritten digits from 0 to 9. There are 60,000 training images and 10,000 test images in the official split. CIFAR-10 consists of 32×32 colour images containing objects in ten generic object categories. This dataset is typically split into 50,000 training and 10,000 test images.

In all of our image experiments, we apply the same sequence autoencoder (SA) architecture as shown in Figure 1a to learn a representation. No pre-processing beyond a global scaling is applied to the MNIST dataset. For CIFAR-10 we apply global normalization and the same crop and flip operations

that Krizhevsky et al. (2012) used for input space data augmentation when training AlexNet (we crop to 24×24). To simulate sequence input the images are fed into the network one row of pixels per time step similar to the SA setup in (Dai & Le, 2015).

For each dataset we train a 2-layer MLP on the context vectors produced by the sequence encoder. Both MLP and SA use the same number of hidden units in each layer: 256 per layer for MNIST and 1024 per layer for CIFAR-10. We conduct four different test scenarios on the MNIST dataset. To control for the representation, as a baseline we trained the classifier only on context vectors from the original images (i.e. SA with no augmentation). We then compare this to training with various kinds of dataset augmentation: traditional affine image transformations in input space (shifting, rotation, scaling), extrapolation between nearest neighbours in input space, and extrapolation between nearest neighbours in representational space. For both extrapolation experiments we use three nearest neighbours per sample and $\gamma = 0.5$ when generating new data. For CIFAR-10, our baseline is trained using context vectors extracted from cropped and flipped images. Against this baseline we test the addition of extrapolation between nearest neighbours in representational space, using the same setup as the MNIST test. Due to the size of the datasets we apply an approximate nearest neighbour algorithm (Wan et al., 2016).

Results are reported in Table 4. For MNIST, we find that extrapolating in feature space not only performs better than the baseline, but it also achieves a lower error rate compared to domain-specific data augmentation in input space. A similar outcome is observed in CIFAR-10, where feature space extrapolation reduces error rate by 0.3%. Interestingly, we note that the baseline test for this dataset already leveraged image transformations to improve performance, so the additional reduction in error rate could indicate that both kinds of augmentation, extrapolation in feature space and manual transformation in pixel space, could complement each other.

Table 4: Test error (%) on MNIST and CIFAR-10. Averages over 10 and 5 runs, respectively.

| Model | MNIST | CIFAR-10 |
|---|---|---|
| Baseline | $1.093 \pm 0.057$ | $32.35 \pm 0.29$ |
| Baseline + input space affine transformations | $1.477 \pm 0.068$ | - |
| Baseline + input space extrapolation | $1.010 \pm 0.065$ | - |
| Baseline + feature space extrapolation | $\mathbf{0.950 \pm 0.036}$ | $\mathbf{31.93 \pm 0.16}$ |

## 5 CONCLUSION

In this paper, we demonstrate a new domain-independent data augmentation technique that can be used to improve performance when training supervised learning models. We train a sequence autoencoder to construct a learned feature space in which we extrapolate between samples. This technique allows us to increase the amount of variability within the dataset, ultimately resulting in a more robust model. We demonstrate our technique quantitatively on five datasets from different domains (speech, sensor processing, motion capture, and images) *using the same simple architecture* and achieve near state-of-the-art results on two of them. Moreover, we show that data augmentation in feature space may complement domain-specific augmentation.

An important finding is that the extrapolation operator, when used in feature space, generated useful synthetic examples while noise and interpolation did not. Additional synthetic data experiments where we could control the complexity of the decision boundary revealed that extrapolation only improved model performance in cases where there were complex class boundaries. In cases with simple class boundaries, such as linear separability or one class encircling another, extrapolation hindered model performance, while interpolation helped. Our current hypothesis is that interpolation tends to tighten class boundaries and unnecessarily increase confidence, leading to overfitting. This behaviour may cause the model to ignore informative extremities that can describe a complex decision boundary and as a result produce an unnecessarily smooth decision boundary. As most high-dimensional, real datasets will typically have complex decision boundaries, we find extrapolation to be well suited for feature space dataset augmentation.

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
