# Peer review of "Dataset Augmentation in Feature Space"

_ICLR 2017 — rejected_

[Official Review · AnonReviewer2 · rating 6 · confidence 5 · 16 Dec 2016]
**No Title**

In this paper authors propose a novel data augmentation scheme where instead of augmenting the input data, they augment intermediate feature representations.  Sequence auto-encoder based features are considered, and random perturbation, feature interpolation, and extrapolation based augmentation are evaluated. On three sequence classification tasks and on MNIST and CIFAR-10, it is shown that augmentation in feature space, specifically extrapolation based augmentation, results in good accuracy gains w.r.t. authors baseline.

My main questions and suggestions for further strengthening the paper are:

a) The proposed data augmentation approach is applied to a learnt auto-encoder based feature space termed ‘context vector’ in the paper.  The context vectors are then augmented and used as input to train classification models. Have the authors considered applying their feature space augmentation idea directly to the classification model during training, and applying it to potentially many layers of the model?  Also, have the authors considered convolutional neural network (CNN) architectures as well for feature space augmentation?  CNNs are now the state-of-the-art in many image and sequence classification task, it would be very valuable to see the impact of the proposed approach in that model.

b) When interpolation or extrapolation based augmentation was being applied, did the authors also consider utilizing nearby samples from competing classes as well?  Especially in case of extrapolation based augmentation it will be interesting to check if the extrapolated features are closer to competing classes than original ones.

c) With random interpolation or nearest neighbor interpolation based augmentation the accuracy seems to degrade pretty consistently.  This is counter-intuitive.  Do the authors have explanation for why the accuracy degraded with interpolation based augmentation?

d) The results on MNIST and CIFAR-10 are inconclusive.  For instance the error rate on CIFAR-10 is well below 10% these days, so I think it is hard to draw conclusions based on error rates above 30%.  For MNIST it is surprising to see that data augmentation in the input space substantially degrades the accuracy (1.093% -> 1.477%).  As mentioned above, I think this will require extending the feature space augmentation idea to CNN based models.

[Official Review · AnonReviewer3 · rating 4 · confidence 5 · 17 Dec 2016]

TDLR: The authors present a regularization method wherein they add noise to some representation space. The paper mainly applies the technique w/ sequence autoencoders (Dai et al., 2015) without the usage of attention (i.e., only using the context vector). Experimental results show improvement from author's baseline on some toy tasks.

=== Augmentation ===
The augmentation process is simple enough, take the seq2seq context vector and add noise/interpolate/extrapolate to it (Section 3.2). This reviewer is very curious whether this process will also work in non seq2seq applications. 

This reviewer would have liked to see comparison with dropout on the context vector.

=== Experiments ===
Since the authors are experimenting w/ seq2seq architectures, its a little bit disappointing they didn't compare it w/ Machine Translation (MT), where there are many published papers to compare to.

The authors did compare their method on several toy datasets (that are less commonly used in DL literature) and MNIST/CIFAR. The authors show improvement over their own baselines on several toy datasets. The improvement on MNIST/CIFAR over the author's baseline seems marginal at best. The author also didn't cite/compare to the baseline published by Dai et al., 2015 for CIFAR -- here they have a much better LSTM baseline of 25% for CIFAR which beats the author's baseline of 32.35% and the author's method of 31.93%.

The experiments would be much more convincing if they did it on seq2seq+MT on say EN-FR or EN-DE. There is almost no excuse why the experiments wasn't run on the MT task, given this is the first application of seq2seq was born from. Even if not MT, then at least the sentiment analysis tasks (IMDB/Rotten Tomatoes) of the Dai et al., 2015 paper which this paper is so heavily based on for the sequence autoencoder.

=== References ===
Something is wrong w/ your references latex setting? Seems like a lot of the conference/journal names are omitted. Additionally, you should update many cites to use the conference/journal name rather than just "arxiv".

Listen, attend and spell (should be Listen, Attend and Spell: A Neural Network for Large Vocabulary Conversational Speech Recognition) -> ICASSP
if citing ICASSP paper above, should also cite Bahandau paper "End-to-End Attention-based Large Vocabulary Speech Recognition" which was published in parallel (also in ICASSP).

Adam: A method for stochastic optimization -> ICLR
Auto-encoding variational bayes -> ICLR
Addressing the rare word problem in neural machine translation -> ACL
Pixel recurrent neural networks -> ICML
A neural conversational model -> ICML Workshop

[Official Review · AnonReviewer1 · rating 7 · confidence 4 · 17 Dec 2016]
**No Title**

The concept of data augmentation in the embedding space is very interesting. The method is well presented and also justified on different tasks such as spoken digits and image recognition etc.

One comments of the comparison is the use of a simple 2-layer MLP as the baseline model throughout all the tasks. It's not clear whether the gains maintain when a more complex baseline model is used. 

Another comment is that the augmented context vectors are used for classification, just wondering how does it compare to using the reconstructed inputs. And furthermore, as in Table 4, both input and feature space extrapolation improves the performance, whether these two are complementary or not?

[Author Response · Terrance DeVries · 25 Jan 2017]
**Update regarding CIFAR-10 Wide ResNet results**

In our previous experiments with the Wide ResNet architecture (see our response to Reviewer1's review for details) we tested to see whether using reconstructed inputs, rather than context vectors, still yielded improved model performance. In these tests we encoded CIFAR-10 images with the encoder portion of a sequence autoencoder, extrapolated between the resulting context vectors, and then projected the context vectors back into input space with the decoder to reconstruct augmented images. The newly generated images were then added to the original dataset. We found that this data augmentation approach resulted in worse classification performance compared to the baseline model. However, we also observed that training a model on only reconstructed images (encoded then decoded, with no extrapolation) yielded results much worse than the equivalent test trained on unmodified data. We hypothesized that the decrease in performance from the tests with extrapolation may possibly be attributed to the decoder's reconstruction error rather than the effects of extrapolation. 

To isolate the effects of extrapolation we conducted several more experiments, this time using reconstructed samples for all tests so that reconstruction error affected all tests equally. Any difference between baseline and extrapolation tests could now be contributed solely to the effects of extrapolation. The results from our tests are shown below:

Test 1 - 24x24 reconstructions, center crop: 18.75 +/- 0.24 (% test error)
Test 2 - 24x24 reconstructions, center crop + extrapolation: 17.72 +/- 0.45 (% test error)
Test 3 - 24x24 reconstructions, simple data augmentation (shift + mirror): 13.55 +/- 0.15 (% test error)
Test 4 - 24x24 reconstructions, simple data augmentation + extrapolation: 11.99 +/- 0.11 (% test error)

In these tests we observed a consistent reduction in classification error when feature space extrapolation is applied: about 1% reduction in test error when applied to 24x24 center crops of the original images, and a 1.5% reduction in test error when coupled with input space data augmentation. We believe that these results demonstrate the potential benefits of the feature space extrapolation technique, which persist even when applied to complex model architectures such as Wide ResNet. As previously mentioned we are currently investigating methods which may reduce the reconstruction error introduced during the decoding process.

[Final Decision · Program Chairs · 06 Feb 2017]
**ICLR committee final decision**

This paper proposes to regularize neural networks by adding synthetic data created by interpolating or extrapolating in an abstract feature space, learning by an autoencoder.
 
 The main idea is sensible, and clearly presented and motivated. Overall this paper is a good contribution. However, the idea seems unlikely to have much impact for two reasons:
  - It's unclear when we should expect this method to help vs hurt
  - Relatedly, the method has a number of hyperparameters that it's unclear how to set except by cross-validation.
 
 We also want to remark that other regularization methods effectively already do closely related things. Dropout, gradient noise, and Bayesian methods, for instance, effectively produce 'synthetic data' in a similar way when the high-level weights of the network are perturbed.